# Genome-Wide Association Analysis for Resistance to *Coniothyrium glycines* Causing Red Leaf Blotch Disease in Soybean

**DOI:** 10.3390/genes14061271

**Published:** 2023-06-15

**Authors:** Musondolya Mathe Lukanda, Isaac Onziga Dramadri, Emmanuel Amponsah Adjei, Arfang Badji, Perpetua Arusei, Hellen Wairimu Gitonga, Peter Wasswa, Richard Edema, Mildred Ochwo-Ssemakula, Phinehas Tukamuhabwa, Harun Murithi Muthuri, Geoffrey Tusiime

**Affiliations:** 1Department of Agricultural Production, College of Agricultural and Environmental Sciences, Makerere University, Kampala P.O. Box 7062, Uganda; lukandamathe6@gmail.com (M.M.L.); emmaadjei1@gmail.com (E.A.A.); dabgarf42@gmail.com (A.B.); arusei.perpetua@gmail.com (P.A.); hellenwairimu@gmail.com (H.W.G.); peterwasswa648@gmail.com (P.W.); redema14@gmail.com (R.E.); mknossemakula@gmail.com (M.O.-S.); tphinehas@yahoo.com (P.T.); gwtusiime@gmail.com (G.T.); 2Makerere Regional Center for Crop Improvement (MaRCCI), Makerere University, Kampala P.O. Box 7062, Uganda; 3Faculté des Sciences Agronomiques, Université Catholique du Graben, Butembo P.O. Box 29, Democratic Republic of the Congo; 4Council for Scientific and Industrial Research-Savanna Agricultural Research Institute, Tamale P.O. Box TL 52, Ghana; 5Department of Biological Sciences, Moi University, Eldoret P.O. Box 3900-30100, Kenya; 6Agricultural Research Service Research Participation Program, Oak Ridge Institute for Science and Education, Oak Ridge, TN 37831, USA; hmurithi@gmail.com; 7International Institute of Tropical Agriculture (IITA), ILRI, Nairobi P.O. Box 30709-00100, Kenya

**Keywords:** *Coniothyrium glycines*, red leaf blotch, soybean, GWAS, resistance

## Abstract

Soybean is a high oil and protein-rich legume with several production constraints. Globally, several fungi, viruses, nematodes, and bacteria cause significant yield losses in soybean. *Coniothyrium glycines* (*CG*), the causal pathogen for red leaf blotch disease, is the least researched and causes severe damage to soybean. The identification of resistant soybean genotypes and mapping of genomic regions associated with resistance to *CG* is critical for developing improved cultivars for sustainable soybean production. This study used single nucleotide polymorphism (SNP) markers generated from a Diversity Arrays Technology (DArT) platform to conduct a genome-wide association (GWAS) analysis of resistance to *CG* using 279 soybean genotypes grown in three environments. A total of 6395 SNPs was used to perform the GWAS applying a multilocus model Fixed and random model Circulating Probability Unification (FarmCPU) with correction of the population structure and a statistical test *p*-value threshold of 5%. A total of 19 significant marker–trait associations for resistance to *CG* were identified on chromosomes 1, 5, 6, 9, 10, 12, 13, 15, 16, 17, 19, and 20. Approximately 113 putative genes associated with significant markers for resistance to red leaf blotch disease were identified across soybean genome. Positional candidate genes associated with significant SNP loci-encoding proteins involved in plant defense responses and that could be associated with soybean defenses against *CG* infection were identified. The results of this study provide valuable insight for further dissection of the genetic architecture of resistance to *CG* in soybean. They also highlight SNP variants and genes useful for genomics-informed selection decisions in the breeding process for improving resistance traits in soybean.

## 1. Introduction

Soybean is a cash crop used for animal feed, human consumption, soil fertility improvement, and industrial use for ethanol and biofuel production [1,2,3]. Soybean is a legume rich in oil and protein content [4]. Despite its importance, soybean production is challenged by several abiotic and biotic constraints [5,6]. Across the world, 26 fungi, 9 viruses, 5 nematodes, and 3 bacteria have been identified to cause diseases in soybean during production [6]. Among biotic constraints, pathogenic fungi have been reported to cause severe foliar diseases that can result in a significant reduction in soybean productivity.

In Africa, many fungal diseases have been reported to reduce soybean productivity [7]. Over the last decades, soybean rust (*Phakopsora pachyrhizi*) has been the major disease because of its economic importance and significant yield losses in soybean growing areas. Since then, red leaf blotch caused by *CG* [8] has expanded and infected soybean in several countries in Africa [5]. In addition to soybean, *CG* infects other legumes including the perennial *Neonotina wightii*, which grows wild in many locations in sub-Saharan Africa [9].

Red leaf blotch disease is a soybean disease native to Africa, and it was first reported in Ethiopia [10]. The disease has continued to spread in many countries in Africa including Zambia and Nigeria, where significant yield losses of over 50% have been reported because of red leaf blotch [11,12,13]. This coupled with other stress has led to many African countries importing soybean to meet the rising market demand [14]. Whereas soybean production has continued to expand [14,15], the rapid spread of red leaf blotch disease is an immense challenge to the soybean sector in Africa. Current predictions indicate that red leaf blotch is likely to cause significant damage to soybean production in many countries including the USA [16,17].

To date, no resistant sources to red leaf blotch have been reported among soybean genotypes [18,19]. Therefore, there is an urgent need to evaluate soybean germplasms for their response to *CG* and to identify markers associated with resistance to accelerate breeding for host resistance in soybean. In addition, soybean genotypes with genes conferring resistance to *CG* would be useful for understanding the mechanisms of resistance to *CG* and to facilitate faster integration of conventional or molecular breeding approaches in soybean breeding programs. Although conventional breeding takes time, a combination of modern breeding tools such as molecular markers with quality phenotyping shortens the breeding cycle for developing new plant varieties [20]; hence, this would allow breeding for host resistance to cope with the increasing expansion of the *CG* disease. 

Recent advances in genomics, bioinformatics, and molecular biology techniques have made it is possible to introgress and track genes of interest including resistance to diseases such as *CG* more efficiently using marker-assisted selection, with great potential to increase soybean production. The introgression of genes for resistance to *CG* into commercial soybean varieties and elite lines will involve identifying new sources of resistance and markers linked to resistance to accelerate the selection process. This also requires good knowledge of how differences in the DNA levels relate to phenotypic differences in the soybean genotypes. 

Genome-wide association Analysis (GWAS) is one of the current methods for identifying genomic regions associated with traits of interest for crop improvement [21,22]. It uses germplasm or populations with diversity richness that have been morphologically characterized and genotyped. GWAS detects genomic regions associated with key traits with higher resolution using markers that are in linkage disequilibrium consisting of association panels and or diversity populations [22]. Compared to QTL studies, GWAS has an advantage in that it can detect smaller chromosomal regions affecting a trait of interest. Furthermore, it provides accurate estimates of the size and direction of the effect of alleles in identified loci [23]. 

Combining phenotypic and genotypic data to identify regions and genes associated with a trait of interest is therefore more accurate compared to other mapping techniques. Hence, this study aimed to identify the genomic regions responsible for resistance to *CG*, the causal agent of the red leaf blotch disease in soybean. The identified genomic regions will be useful to the soybean breeding program in the marker-assisted selection for resistance to *CG*. This study provides information for the exploration of disease resistance-related genes and lays a foundation for genetic improvement and variety breeding for soybean resistance to *CG*. 

## 2. Materials and Methods

### 2.1. Planting Materials 

Two hundred seventy-nine (279) soybean genotypes were collected from the National Crops Resources Research Institute (NaCRRI) at Namulonge–Uganda and Makerere University, with sources from four different countries including Uganda, China, the USA, and Zimbabwe (Table 1). Appendix A provides details on the genetic materials used in this study. 

### 2.2. Experimental Design

The 279 soybean genotypes were planted in a randomized complete block design (RCBD) with 2 replications at the Makerere University Agricultural Research Institute Kabanyoro (MUARIK) for two seasons (2021–2022) and at the Nakabango District farm in Jinja–Uganda for the 2022 season. The experimental sites are known as hotspot areas for red leaf blotch disease in soybean in Uganda. Twenty seeds were sown in a line plot of 1 m, with a spacing from other plots of 0.6 m. Weeding was performed by hand twice at 20 and 55 days after planting. 

### 2.3. Scoring of Soybean Red Leaf Blotch

Red leaf blotch disease was recorded seven times in a panel of varying soybean genotypes from 30 days after planting up to reproductive stage 6 (R6). The stages R4 to R6 are recognized as the best fit for disease scoring in soybean production [24,25]. The severity of the red leaf blotch disease was evaluated using a 0–5 scale, which has previously been used to score red leaf blotch disease in soybean [19,26,27]. This scale is based on the observed damage, such as the percentage of the leaf area affected, fragmentation of the leaf, presence of pycnidia on blotches, and the color and size of the blotches (Table 2). 

### 2.4. Genotyping and Quality Control

Fresh leaves were collected and kept on three 96-well plates at 15 days after germination. The three plates were expedited to the Integrated Genotyping Service and Support (IGSS) of the Biosciences in Eastern and Central Africa—ILRI Hub, Kenya, for genotyping. The DNA was extracted from the leaf tissues using the Nucleomag Plant Genomic DNA extraction kit [28], and the DNA quality check was conducted on 0.8% agarose. Genotyping was performed using Diversity Array Technology sequencing (DArTseq). Then, a genomic DNA library was constructed using genomic complexity reduction technology [29]. The library was purified and quantified for cluster generation in an automated clonal amplification system (cBOT Illumina). Thereafter, next-generation sequencing was performed using the sequencer HiSeq 2500 (Illumina, San Diego, CA, USA). 

## 3. Statistical Analysis

### 3.1. Phenotyping Analysis

Phenotypic data obtained from the three environments were pooled and subjected to a linear mixed model analysis using the lme4 package implemented in R. The best linear unbiased estimates (BLUEs) for three environments were obtained by considering the genotypes’ effects as fixed and the environment and replication effects as random in the mixed model as follows: Y*_ijk_*= μ + B*_i_* + G*_j_* + E*_k_* + GE*_jk_* + ε*_ijk_*
where Y*_ijk_* = phenotypic observation for a trait, μ = grand mean, E = environment effect, B = replication effect, G = genotype effect, GE = genotype by environment interaction, and ε = random residual error. 

### 3.2. GWAS Analysis, Genes Annotation, and Linkage Disequilibrium

To perform the GWAS, a multilocus model Fixed and random model Circulating Probability Unification (FarmCPU) with correction of the population structure and a statistical test *p*-value threshold of 5% was used [30]. The Manhattan and quantile–quantile (QQ) plots were plotted using the R package “*rMVP*” (a memory-efficient, visualization-enhanced, and parallel-accelerated tool for genome-wide association study) [31]. 

The SNP markers significantly associated with resistance to the red leaf blotch disease identified through GWAS were annotated using the Phytozome 13.0 database (https://phytozome-next.jgi.doe.gov/info/Gmax_Wm82_a4_v1, accessed on 10 December 2022) and used as the source for the candidate gene search. The linkage disequilibrium (LD) was estimated among the significant SNPs using the “*LDheatmap*” library [32]. The LD decay rate of 90 to 574 kb has been reported in soybean [33] and a ±574 kb region was used to identify positional candidate genes [34]. 

In selecting candidate genes, the following criteria [34,35] were used as (i) genes of known function in soybean related to the trait under study, (ii) genes with function-known orthologs in Arabidopsis related to the trait under study, and (iii) genes pinpointed by the peak SNPs. The public database InterPro, European Molecular Biology Laboratory–European Bioinformatics Institute (EMBL-EBI), was used to determine the functions of the genes associated with the different SNPs identified [36]. 

The SNPs’ contributions to the resistance to red leaf blotch disease based on the observed alleles were plotted using the “ggplots”, and their confidence statistics were calculated with “rstatix” in R.

## 4. Results

### 4.1. Phenotypic Variation 

In this study, significant (*p* < 0.001) differences were observed among the soybean genotypes for their response to the red leaf blotch disease under natural infestation. The analysis of variance (ANOVA) revealed that genotypes, environments, and genotypes by environment interaction were the main sources of variation (Table 3). Out of the 279 soybean genotypes, approximately 10.75% were evaluated as resistant to red leaf blotch disease (Figure 1). Figure 1 shows the resistant genotypes, represented by the dots on the bottom. The heritability values observed from the studied traits ranged from 0.23 to 0.70, with the lowest heritability identified for R_1_ (Table 3).

**Table 3 genes-14-01271-t003:** Analysis of the variance for the response to red leaf blotch disease under natural infestation of 279 soybean genotypes at different scoring moments.

Source of Variation	DF ^a^	R_1_ ^b^	R_2_	R_3_	R_4_	R_5_	R_6_
Rep	1	0.538	0.263	4.12	0.86	1.32	1.49
Genotypes	278	0.177	2.453 ***	3.67 ***	4.92 ***	5.8 ***	6.56 ***
Environment	2	12.836 ***	21.57 ***	39.32 ***	85.86 ***	86.58 ***	84.05 ***
Geno * Enviro	556	0.141	0.761 ***	1.21 ***	1.54 ***	1.84 ***	2 ***
Residuals	836	0.112	0.37	0.38	0.51	0.59	0.69
Grand Mean		0.27	1.09	1.25	1.49	1.63	1.73
LSD		0.29	0.82	1.02	1.17	1.27	1.34
CV		142.11	53.12	46.90	46.43	45.63	45.95
Heritability		0.23	0.69	0.67	0.69	0.68	0.70

^a^ Degree of freedom. ^b^ Soybean reproductive stage. Significance level: * <0.05; *** <0.001.

**Figure 1 genes-14-01271-f001:**
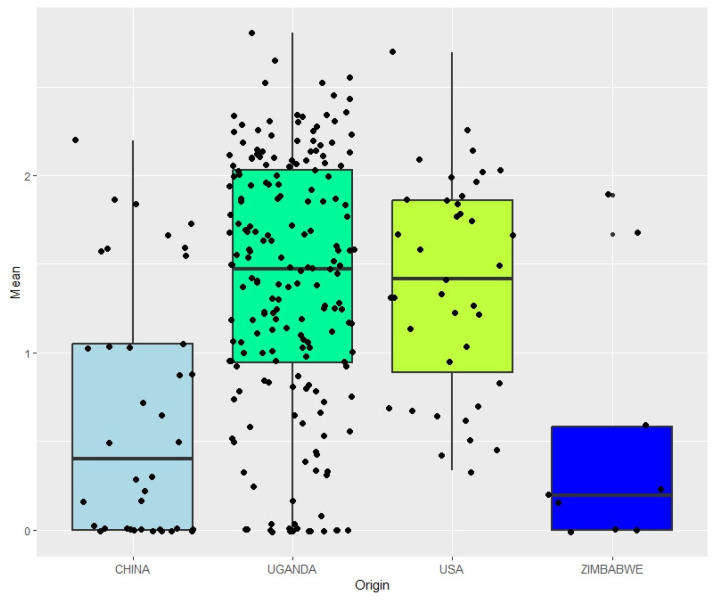
Soybean genotypes resistant to red leaf blotch disease and their origin. Dots on the bottom represent resistant soybean accessions. The *x*-axis represents the origin, and the *y*-axis represents the general mean of the disease scores. Additional information concerning the genotypes is available in Appendix A.

### 4.2. Marker Coverage and Distribution 

The number of Diversity Array Technology sequencing (DArTseq)-generated SNP markers was 14,082. A large number (7687) were discarded after the filtering and imputation of the raw data, and the remaining markers was 6395 SNPs, approximately 45.41% of the DArTseq-generated SNP markers. The 6395 SNPs markers matched the criteria of the data for use in the GWAS. The 6395 SNPs were distributed across the 20 *Glycine max* chromosomes. Chromosome 12 and chromosome 18 have, respectively, a small (201) and high (476) number of SNPs (Figure 2, Table 4). The MAF (minor allele frequency) and PIC (polymorphism information content) of these SNPs ranged from 0.043 to 0.5, with an average of 0.22, and 0.08 to 0.74, with an average of 0.29, respectively.

**Table 4 genes-14-01271-t004:** Chromosomes size and number of SNPs for *Glycine max* chromosomes after filtering alongside the average polymorphism information content.

Chr ^a^	FSNPs ^b^	MAF ^c^	GD ^d^	PIC ^e^
1	239	0.144	0.37	0.30
2	353	0.143	0.33	0.28
3	298	0.045	0.35	0.30
4	291	0.144	0.37	0.30
5	257	0.171	0.37	0.31
6	402	0.143	0.32	0.27
7	285	0.125	0.33	0.28
8	330	0.137	0.34	0.29
9	370	0.166	0.37	0.31
10	276	0.125	0.33	0.28
11	235	0.114	0.33	0.28
12	201	0.133	0.35	0.30
13	362	0.147	0.36	0.30
14	346	0.135	0.32	0.27
15	341	0.159	0.38	0.32
16	348	0.138	0.34	0.29
17	351	0.147	0.33	0.28
18	476	0.152	0.36	0.30
19	332	0.153	0.37	0.31
20	302	0.113	0.33	0.28
Total/Average	6395	0.783	0.348	0.293

^a^ Chromosome. ^b^ Filtered single nucleotide polymorphism. ^c^ Minor allele frequency. ^d^ Gene diversity. ^e^ Polymorphism information content.

**Figure 2 genes-14-01271-f002:**
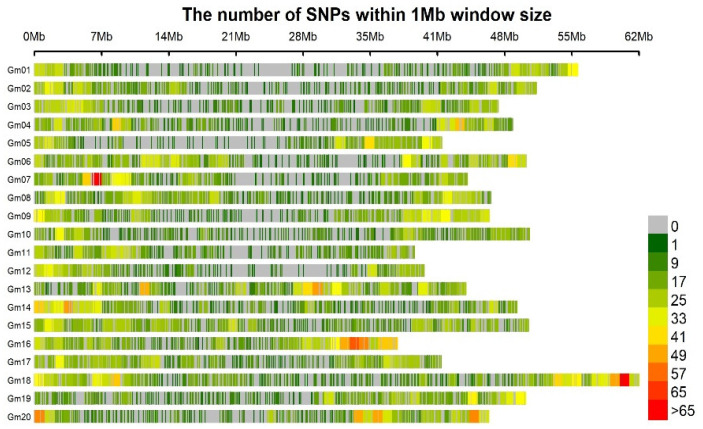
Genome-wide SNP coverage showing the number of SNPs within a 1 Mb window size. Chromosomes appear horizontally with the density of the SNPs depicted in the scale shown to the right.

### 4.3. Association Results

The FarmCPU model was used in this study to reveal the loci associated with the resistance to *CG* causing the red leaf blotch disease of soybean. In total, 19 significant association signals were found at various reproductive stages of the crop growth (Table 5). At R_1_, four signals were identified on chromosomes 1, 13, 15, and 16 (Table 5, Figure 3A). At R_2_, four signals were identified on chromosomes 12, 16, 17, and 19 (Table 5, Figure 3B). At R_3_, two signals were identified on chromosome 6 (Table 5, Figure 3C). At R_4_, five signals were identified on chromosomes 1, 6, 16, and 20 (Table 5, Figure 3D). At R_5_, four signals were identified on chromosomes 5, 6, 16, 17, and 19 (Table 5, Figure 3E). At R_6_, five signals were identified on chromosomes 1, 6, 9, 10m and 19 (Table 5, Figure 3F). The quantile–quantile plots (QQ plots) produced by displaying the negative logarithms (−log_10_) for the *p*-values against their *p*-values demonstrated that the genome-wide association study analysis model was reasonable in this research. Differences between observed and expected values of the traits studied in this research were identified, and they indicate a link between the phenotypic and SNPs, as demonstrated by the QQ plots (Figure 3).

**Table 5 genes-14-01271-t005:** Summary of significant SNPs associated with resistance to *Coniothyrium glycines* evaluated at six soybean reproductive stages (R1-6) in the germplasm of 279 soybean genotypes.

RS ^a^	SNPs ID ^b^	CHR ^c^	POS ^d^	Alleles	Effect	SE
R_1_	Gm01_36009335	Gm01	36,009,335	A/T	−0.017	0.004
	Gm13_40079851	Gm13	40,079,851	A/T	0.022	0.005
	Gm15_12688260	Gm15	12,688,260	T/A	0.022	0.005
	Gm16_3302971	Gm16	3,302,971	A/G	0.037	0.009
R_2_	Gm12_34424219	Gm12	34,424,219	G/G	−0.104	0.024
	Gm16_34649045	Gm16	34,649,045	G/A	−0.213	0.048
	Gm17_8014133	Gm17	8,014,133	T/C	0.132	0.026
	Gm19_35502386	Gm19	35,502,386	G/T	0.148	0.034
R_3_	Gm06_20112134	Gm06	20,112,134	A/G	−0.320	0.034
	Gm06_38404808	Gm06	38,404,808	A/G	−0.259	0.056
R_4_	Gm01_17813710	Gm01	17,813,710	A/G	−0.183	0.061
	Gm06_20112134	Gm06	20,112,134	A/G	−0.318	0.045
	Gm16_3302971	Gm16	3,302,971	A/G	0.472	0.086
	Gm16_34649045	Gm16	34,649,045	G/T	0.489	0.122
	Gm20_34576213	Gm20	34,576,213	G/T	0.209	0.051
R_5_	Gm05_30968142	Gm05	30,968,142	A/C	0.135	0.034
	Gm06_20112134	Gm06	20,112,134	A/G	−0.450	0.044
	Gm16_31759458	Gm16	31,759,458	C/T	0.789	0.190
	Gm17_14222127	Gm17	14,222,127	T/C	0.265	0.059
	Gm19_44916522	Gm19	44,916,522	T/C	0.238	0.037
R_6_	Gm01_17813710	Gm01	17,813,710	A/G	−0.222	0.072
	Gm06_19862041	Gm06	19,862,041	G/A	−0.620	0.047
	Gm09_4708504	Gm09	4,708,504	G/A	0.476	0.104
	Gm10_48178692	Gm10	48,178,692	C/G	0.150	0.041
	Gm19_44916522	Gm19	44,916,522	T/C	0.261	0.040

^a^ Reproductive stage for disease scoring. ^b^ Single nucleotide polymorphism. ^c^ Chromosome number. ^d^ Position.

Some SNPs were significant at more than one scoring time. This was the case for Gm01_17813710 at R_4_ and R_6_; Gm06_20112134 at R_3_, R_4_, and R_5_; Gm16_3302971 at R_1_ and R_4_; Gm16_34649045 at R_2_ and R_4_; and Gm19_44916522 at R_5_ and R_6_ (Table 5). The Manhattan plot reveals the results of the GWAS significance levels (−log_10_ of the *p*-value of each SNP) by chromosome position, where each chromosome has a different color. Significant SNPs in the Manhattan plot are strongly associated with resistance to red leaf blotch disease in soybean (Figure 3).

**Figure 3 genes-14-01271-f003:**
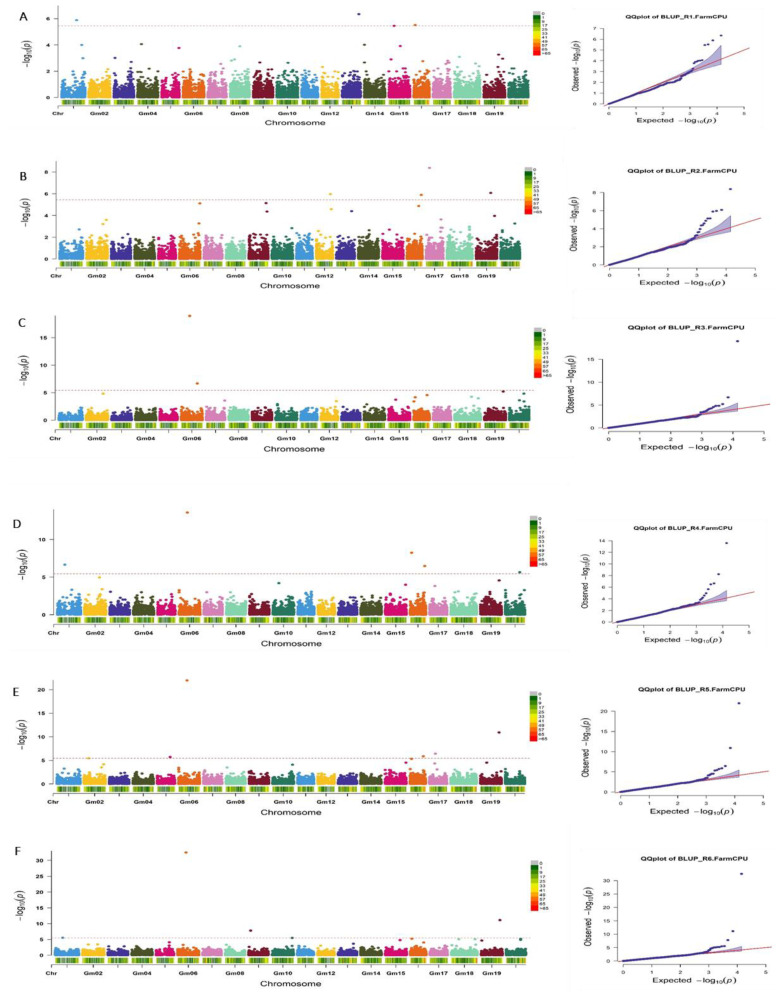
Manhattan plots for the genome-wide diagnosis of association signals for resistance to red leaf blotch disease in soybean (**left**) and quantile–quantile (QQ) plots of the *p*-values (**right**). (**A**–**F**) reproductive stages 1, 2, 3, 4, 5, and 6, respectively. Manhattan plots: The *x*-axis is the genomic position of the SNPs in the genome, and the *y*-axis is the negative log base 10 of the *p*-values. The red horizontal line indicates the significance level. QQ plot: the *y*-axis is the observed negative base 10 logarithm of the *p*-values, and the *x*-axis is the expected observed negative base 10 logarithm of the *p*-values.

### 4.4. Genes Identity, Discovery, and Annotations

Positional candidate genes associated with the significant SNP markers were identified using soybean reference genome Gmax_Wm82_a4_v1 available in Phytozome 13.0 (https://phytozome-next.jgi.doe.gov/info/Gmax_Wm82_a4_v1 (accessed on 9 March 2023)). This resulted in the discovery of 113 candidate genes that encode for proteins that were characterized. Functional annotation of these proteins suggests their involvement in the plant’s growth and response to various abiotic and biotic stress. Several candidate genes associated with significant markers for resistance to red leaf blotch disease were distributed in the soybean genome on chromosomes 1 (6 genes), 5 (5 genes), 6 (19 genes), 9 (6 genes), 10 (7 genes), 12 (5 genes), 13 (7 genes), 15 (7 genes), 16 (17 genes), 17 (11 genes), 19 (14 genes), and 20 (9 genes) (Table 6).

### 4.5. SNPs’ Contribution to the Resistance to Red Leaf Blotch Disease in Soybean

Five common significant SNPs were identified for at least two scoring times including Gm01_17813710 at R4 and R6; Gm06_20112134 at R3, R4, and R5; Gm16_3302971 at R1 and R4; Gm16_34649045 at R2 and R4; and Gm19_44916522 at R5 and R6. Further dissection of the five significant SNP loci associated with resistance to red leaf blotch showed that accessions with the homozygous allele AA and/or heterozygous allele AG on chromosome 1 possessed higher resistance than the homozygous allele GG (Figure 4A,B). On chromosome 6, resistance is associated with the homozygous allele GG (Figure 4C–E). The marker effect on chromosomes 16 and 19 revealed that, respectively, the homozygous alleles GG and TT (Figure 4F–I) and CC and TT (Figure 4J,K) are linked with resistance in the studied population, while the heterozygous allele accounted for low resistance. The SNP confidence statistics revealed an allelic significative difference (*p* < 0.001) at SNPs Gm01_17813710 and Gm06_20112134 (Figure 4A–E). At *p* < 0.001, the other SNPs did not show an allelic significative difference (Figure 4).

In general, resistance to red leaf blotch disease in soybean is characterized by homologous allele GG at positions Gm06_20112134 and Gm16_34649045. The position Gm16_3302971 is dominated by the homologous allele TT, except for two soybean genotypes (UGSOY143 and UGSOY236) that are characterized by the heterozygous allele TA. The position Gm19_44916522 is predominated by homologous alleles CC or TT, but one genotype (UGSOY195) has a heterozygous allele TC (Table 7, Figure 4)

### 4.6. Haplotype Analysis

Linkage disequilibrium (LD) block heatmaps based on the LD of each identified common SNP loci are shown in Figure 5. The LD analysis of the common loci (two on chromosomes 1 and 19 and three on chromosomes 6 and 16) showed that these markers had a relatively average to high LD parameter (R^2^ > 0.8), showing a relatively high correlation.

## 5. Discussion

Identifying novel sources of resistance in soybean germplasm to key biotic and abiotic stress is an essential determinant for enhancing productivity [91,92]. Extensive work on the genetic improvement of soybean to resistance to selected diseases and pests [92] and other nutritional value [93] has been conducted across the world, but limited efforts have been made in the area of red leaf blotch disease in soybean [26]. In this study, a GWAS was conducted to map genomic regions associated with *CG* resistance. A total of 19 significant GWAS signals were reported for resistance to red leaf blotch disease in soybean. In addition, putative candidate genes associated with resistance were also identified. The markers identified in this study provide a means to accelerate the development of soybean cultivars with resistance and with other acceptable end-user attributes. 

The ability of a GWAS to dissect complex traits has been demonstrated in soybean quality improvement and breeding for seed composition [37] and for resistance to soybean rust [38]. The consumption of soybeans is increasing and, consequently, breeding programs need to be encouraged and optimized with new knowledge. The current GWAS was intended for the discovery of QTLs and potential candidate genes linked to genetic diversity for resistance to red leaf blotch disease in soybean. The knowledge of the population structure and familial relationships (i.e., kinship) in an association panel is important to prevent false associations in a GWAS [39]. Population structure and admixture for this population were reported in a previous study [40]. From our study, 113 putative candidate genes encoding for several proteins (Table 6) were discovered as major contributors to the resistance to *Coniothyrium glycines*, the causal agent of red leaf blotch disease in soybean production. 

Squamosa promoter-binding protein-like (SPL) genes play vital regulatory roles in plant growth, development, and stress responses [41,42] and show potential application in crop improvement by genetic modification for abiotic stress in Alfalfa (*Medicago sativa* L.) [43]. SPL genes are reported to play a role in toxin resistance in plants [44]. Plastid lipid-associated proteins, also termed fibrillin, are known for their role in response to biotic stress in Solanaceae plants, especially for bacterial infections [45]. Isochorismate synthase 2 contributes to the biosynthesis of salicylic acid [46], which is involved in plant defenses [47]. In soybean, the defense responses to the pathogens *Pseudomonas syringae* and *Phytophthora sojae* is conferred by the accumulation of salicylic acid [48]. Proteasome is reported to contribute to the tolerance of heat or oxidative stresses in plants [49,50]. The RING/U-box superfamily protein promotes resistance to biotic stress through ubiquitination and leaf senescence [51]. The U-box protein is known to play a major role in responses to abiotic and biotic stresses in rice [52] and regulates drought tolerance in *Arabidopsis thaliana* [53]. The action of the RHO guanyl-nucleotide exchange factor 7 was demonstrated in the development, pathogenesis, and stress responses of *Colletotrichum higginsianum*, which causes anthracnose disease of crucifers [54]

The chaperone DnaJ-domain superfamily protein has been reported in pepper (*Capsicum annuum* L.) to play a role in plant growth and development and heat stress [55]. Hence, it is called heat shock protein 40 based on molecular weight [55]. In wheat, the chaperone DnaJ-domain is reported in the regulation of resistance to yellow mosaic virus infection [56]. Earlier, in 2013, a critical role of the nuclear-localized DnaJ domain-containing GmHSP40.1 in cell death and disease resistance in soybean was demonstrated through the screening for candidate genes stimulating cell death in soybean, and silencing GmHSP40.1 enhanced the susceptibility of soybean plants to soybean mosaic virus, confirming its positive role in pathogen defense [57]. Another positive role of the DnaJ-domain is that it is involved in the alkaline-salt, salt, and drought tolerance in Arabidopsis and soybean [58]. The chaperone DnaJ-domain plays a critical function in protein folding and regulation of several physiological processes, and it participates in numerous pathological processes [59]. DnaJ-domain superfamily proteins have been recognized for their diverse functions within cells and extensively studied in many species, including humans, drosophila, Arabidopsis, mushrooms, and tomatoes [60]. Furthermore, this information on the role played by the chaperone DnaJ-domain may guide practical actions in soybean breeding for resistance to red leaf blotch disease. 

Enolase is involved in the growth and development of various species [61]. In soybean, the flooding stress is controlled by enolase with the contribution of other proteins [62]. UDP-glucosyl transferase 89B1 plays a vital role in diverse plant functions, and its response to drought, salt, and heat stress in *Populus trichocarpa* (Black cottonwood) has been revealed [63]. The potential role of the EF-hand calcium-binding protein family in the implementation of resistance to environmental and nutritional stress in soybean was described [64]. This calcium-dependent protein has a function in the soybean–herbivore insect interaction and in drought adaptation [65]. Glycine-rich protein-containing protein-like confer tolerance to stress (e.g., some are involved in cold acclimation and may improve growth at low temperatures), and these proteins could play a promising role in agriculture [66]. 

AP2/B3-like transcriptional factor family proteins were analyzed for their role in stress tolerance in soybean [67], tree plants [68,69,70], and *A. thaliana* [71]. A member of the AP2/ERF transcription factor family, *GmERF3*, was isolated from soybean. There was an enhanced resistance against infection by *Ralstonia solanacearum*, *Alternaria alternata,* and tobacco mosaic virus, as well as tolerance to high salinity and dehydration stresses in transgenic tobacco plants [72]. The tetratricopeptide repeat (TPR)-containing protein has functions in plant hormone signaling, and the protein TTL1, containing TPR motifs, is required for abscisic acid responses and osmotic stress tolerance in plants [73,74]. 

The Transducin/WD40 repeat-like superfamily protein is a functional group that has been reported in plant cell wall formation [75]. In soybean, Transducin/WD40 repeat-like proteins were reported to putatively control the total number of flower and pods [76], while in Arabidopsis, they control seed germination, growth, and biomass accumulation [89]. A gene that encodes for Transducin/WD40 repeat-like proteins were identified in wheat and associated with plant tolerance to abiotic stresses [77]. The genes that encode for Transducin/WD40 repeat-like superfamily protein in soybean may play a role in controlling red leaf blotch by rehabilitating the cells damaged by the pathogen during the employment of the disease epidemic’s mechanism. Therefore, the soybean genotypes that present the QTLs involving Transducin/WD40 repeat-like superfamily protein production could be used as future resources for breeding efforts aimed at improving resistance to red leaf blotch disease. 

The leucine-rich repeat (LRR) family protein is well-known for controlling disease resistance in crops, including soybean [78]. In soybean, LRR has been reported to regulate the immune response to Phytophthora root rot [79], coordinate the responses against root-lesion nematode [80], and mediate the response to soybean mosaic virus [81]. This study also revealed the contribution of the LRR domain in the resistance to red leaf blotch disease in soybean production. Phytochrome-associated protein 2 is crucial in photoperiod adaptability and, therefore, influences the flowering time. Liu et al., (2008) showed that phytochromes contribute to the establishment of an adaptive response of soybean to environments, and thus the role of contributing to the resistance to red leaf blotch disease in soybean was revealed. The role of malic enzymes in plant growth and response to stress is documented and mainly discovered in cytoplasmic stroma, mitochondria, and chloroplasts. Previous studies have shown that malic enzymes participate in the process of coping with drought, high salt, and high temperature by increasing water use efficiency and improving photosynthesis by plants [83]. Through the improvement of photosynthesis, malic enzymes contribute to the resistance to red leaf blotch disease.

Various studies have shown that ubiquitination plays a key role in stress response and yield constitution [87]. In the UniProtKB database, 2429 ubiquitin-related proteins are predicted in soybean [85]. Ubiquitin plays a key role in regulating the resistance of soybean to *Heterodera glycines*, a soybean cyst nematode causal agent [84]; *P. sojae*, an infection that causes stem and root rot [86]; and heat shock [87]. In soybean and other legumes, the C2H2-type zinc finger protein is reported to enhance legume–rhizobia symbiosis [88], which is a key physiological process that can limit nitrogen in plants, affecting their growth and development [90]. The accumulation of isoflavone in soybean is governed by the C2H2-type zinc finger protein [94,95], and isoflavone contributes to human health and plant stress tolerance [96]. The soybean C2H2-type zinc finger protein with a conserved QALGGH motif negatively regulates drought responses [97], but it was reported to enhance tolerance to cold [98] in transgenic Arabidopsis. Although several genes that encode for the C2H2-type zinc finger protein have been reported to play various roles in the life mechanism of soybean [95,97,98], the accumulation of unfolded proteins such as the homolog of mammalian P58IPK in the endoplasmic reticulum have built up a conserved mechanism that regulates the stress response in this cell part [99]. The endoplasmic reticulum stress response plays an important role that allows plants to sense and respond to adverse environmental conditions, such as heat stress, salt stress, and pathogen infection [99,100]. These unfolded proteins were revealed, for the first time, as factors that contribute to resistance to red leaf blotch disease in soybean. The role of FASCICLIN-like arabinogalactan-protein in the response to plant pathogens was elucidated by Wu et al., (2020). FASCICLIN-like arabinogalactan-protein controls the infection with the turnip mosaic virus and *P. syringae pv tomato* strain DC3000 (*Pst* DC3000) in *Nicotiana benthamiana,* which is a model plant to study plant–pathogen interactions [101]. Recently, their contribution to resistance to clubroot disease stress in *Brassica napus* was characterized [102]. FASCICLIN-like arabinogalactan-protein was reported to be associated with soluble sugar content in vegetable soybean [103]. The thioredoxin superfamily protein with glutoredoxin regulates the response of nodulated soybean plants to water-deficit stress [104], and it is involved in several plant life mechanisms, including adaptation to environmental stresses [105] or signaling plant immunity [106]. 

The NRAMP metal ion transporter family protein plays a key role in nodule iron homeostasis to support bacterial nitrogen fixation in soybean production [107] and contributes to avoiding cadmium toxicity [108]. The PLATZ transcription factor family protein increases drought tolerance in soybean hairy roots [109]. The CLAVATA3/ESR-RELATED 9 is a key component that modulates the effect of infection with plant-parasitic nematodes [110]; therefore, CLAVATA3/ESR-RELATED 9 interacts with the nematodes in the process of the establishment of feeding sites on the plant roots. WRKY DNA-binding protein 3 is well known in soybean to promote resistance to cyst nematodes [111]. Soybean WRKY-type transcription factor genes GmWRKY13, GmWRKY21, and GmWRKY54 confer differential tolerance to abiotic stresses in transgenic Arabidopsis plants [82]. Therefore, the potential role of WRKY DNA-binding protein in resistance to red leaf blotch is irrefutable. The cell wall/vacuolar inhibitor of fructosidase 1 was reported to regulate the abscisic acid response and salt tolerance in Arabidopsis [111]. The salinity symptoms in plants are characterized by leaf discoloration and damage. In fact, red leaf blotch disease affects the leaf; therefore, the cell wall/vacuolar inhibitor of fructosidase 1 may play a key role in reducing the expansion of leaves’ damage. 

Regarding the analysis and functions played in plants by the proteins encoded by the 113 unrevealed genes of soybean, these genes are directly or indirectly responsible for resistance to *C. glycines* causing red leaf blotch disease in soybean production. These genes govern, in a mutual way, the physiological activities of the soybean plant to enhance resistance to *C. glycines* infection. 

Information on the markers’ effects through segregation patterns are fundamental for conversion of the markers to Kompetitive Allele-Specific PCR (KASP) for genotyping of the polymorphisms at different loci and deployment in breeding programs [112,113] for resistance to red leaf blotch disease in soybean. Some of the markers, especially marker Gm02_17813710 at R4 and R6, have the GG and GA alleles, which significantly reduce disease symptoms compared to the second homozygous state. The same pattern was observed for Gm16_3302971 on R1 and R4, with homozygous state TT and heterozygous state AT, which are superior to that of the homozygous allele AA in minimizing the disease severity. These SNPs demonstrate a dominance effect, and both the favorable homozygous and heterozygous alleles can be exploited for KASP markers in breeding for resistance. Marker Gm06_20112134, on the other hand, shows no significant difference in the allelic effect between AA and GG, while both homozygous states are superior to the heterozygous alternative. This SNP allelic effect pattern may encourage use of both homozygous alleles for KASP marker development. However, its effect on disease needs to be further investigated, for example, through direct molecular analyses and intervention, to ascertain their influence on disease severity. Possible reasons for such allelic behavior could be genetic, such as genetic heterogeneity or statistical, for instance, a low sample size and statistical power in the detection of the SNP effect. SNP Gm19_44916522 shows no differences in the allelic effect among its three allelic states, which suggests that it could be a false positive or affected by allelic heterogeneity, which prompts the necessity to further investigate and validate these alleles.

## 6. Conclusions

This study identified 19 significant markers and genomic regions associated with red leaf blotch disease resistance in soybean. These markers tagging key genomic regions can be validated and tested in the soybean germplasm. This might be performed by transforming these significant markers to low-cost Kompetitive Allele-Specific PCR (KASPs) markers capable of being used effectively to transfer alleles into elite soybean genotypes for use in future marker-based breeding strategies. The findings of this study will contribute to the implementation of a new approach to soybean breeding for maintaining red leaf blotch disease resistance. The discovered genes from this study provide new insight into the genetic foundation of resistance to red leaf blotch disease in genetic pool of soybean. 

## Figures and Tables

**Figure 4 genes-14-01271-f004:**
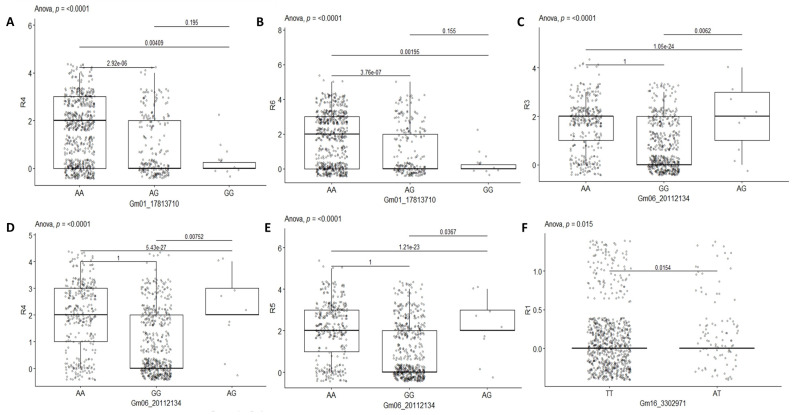
Boxplots showing the effect of the significant markers associated with resistance to red leaf blotch on chromosome 1 (**A**,**B**), chromosome 6 (**C**–**E**), chromosome 16 (**F**–**I**), and chromosome 19 (**J**,**K**). The letters on the *x*-axis represent allele variants (A, C, G, and T).

**Figure 5 genes-14-01271-f005:**
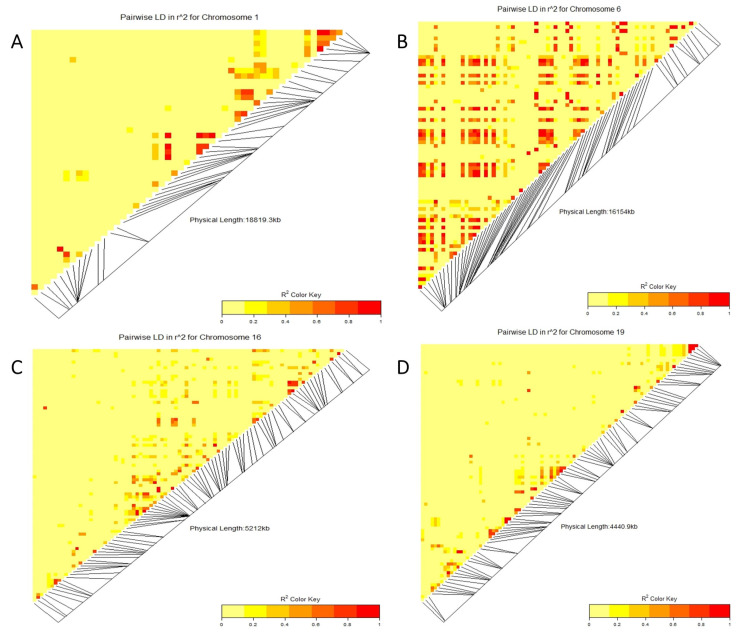
Linkage disequilibrium (LD heatmap) showing the pairwise LD among the SNP markers covering entirely chromosome 1 (**A**), 6 (**B**), 16 (**C**), and 19 (**D**), carrying the genes encoding resistance to red leaf blotch. The red color shows the markers with high LD followed by yellow.

**Table 1 genes-14-01271-t001:** Source, quantity, and summary description of the 279 soybean genotypes.

Origin	Quantity	Description
Uganda	192	Variable in height, number of days to maturity, and pubescence colors.
China	38	Variable in seed color, height, days to flowering, and seed size.
USA	40	Variable in leaf size and shape, plant color, and seeds.
Zimbabwe	9	Variable in lodging, height, seeds, and days to maturity.

**Table 2 genes-14-01271-t002:** Scale to assess the severity of red leaf blotch disease in soybean production.

Category	Description
0	No visible symptoms.
1	Few to many tiny purple–maroon spots (up to 2 mm in diameter), covering 1 to 10% of the leaf area.
2	Mainly large purple–maroon spots (up to 5 mm in diameter), covering 11 to 35% of the leaf area.
3	Purple–maroon or brown blotches (up to 10 mm in diameter) without pycnidia, covering 36 to 65% of the leaf area.
4	Dark brown blotches with pale, bleached centers and pycnidia present, covering 66 to 90% of the leaf, with fragmentation of the leaf starting.
5	91 to 100% of the leaf area affected, extensive blotching and fragmentation of the leaf.

**Table 6 genes-14-01271-t006:** Annotation for the significant SNPs associated with resistance to red leaf blotch disease in soybean.

SNP ID ^a^	Chr ^b^	Gene_ID ^c^	Function	Pfam ^d^	References
Gm01_17813710	1	Glyma.01G075900	Squamosa promoter-binding protein-like 8	PF03110	[37]
		Glyma.01G076100	Plastid lipid-associated protein PAP/fibrillin family protein	PF04755	[38]
Gm01_36009335	1	Glyma.01G104100	Isochorismate synthase 2	PF00425	[39]
		Glyma.01G105500	Proteasome subunit PAB1	PF10584, PF00227	[40]
		Glyma.01G105900	RHO guanyl-nucleotide exchange factor 7	PF03759	[41]
		Glyma.01G106000	Glutathione S-transferase TAU 8	PF00043, PF13417	[42]
Gm05_30968142	5	Glyma.05G115700	RING domain ligase 2	PF13920, PF07002	[43]
		Glyma.05G117233	Alternative oxidase 2	PF01786	[44]
		Glyma.05G117500	K+ uptake permease 7	PF02705	[45]
		Glyma.05G118200	Pumilio 7	PF00806	[46]
		Glyma.05G119600	BRI1-associated receptor kinase	PF08263, PF00069	[47]
Gm06_19862041	6	Glyma.06G207701	Chaperone DnaJ-domain superfamily protein	IPR036869	[48,49]
		Glyma.06G206800	Transducin family protein/WD-40 repeat family protein	PF00400, PF12341	[50]
		Glyma.06G208200	Enolase 1	PF03952 PF00113	[51]
		Glyma.06G206700	UDP-glucosyl transferase 89B1	PF00201	[52]
		Glyma.06G208800	EF-hand calcium-binding protein family	PF13499	[53]
		Glyma.06G207900	Glycine-rich protein-containing protein-like	PF07173	[54]
		Glyma.06G207800	AP2/B3-like transcriptional factor family protein	IPR017392	[55]
		Glyma.06G207751	Tetratricopeptide repeat (TPR)-containing protein	PF13236	[56]
Gm06_20112134	6	Glyma.06G207701	Chaperone DnaJ-domain superfamily protein	IPR036869	[48]
		Glyma.06G208200	Enolase 1	PF03952, PF00113	[51]
		Glyma.06G206700	UDP-glucosyl transferase 89B1	PF00201	[52]
		Glyma.06G207800	AP2/B3-like transcriptional factor family protein	IPR017392	[55]
		Glyma.06G207751	Tetratricopeptide repeat (TPR)-containing protein	PF13236	[56]
		Glyma.06G208800	EF hand calcium-binding protein family	PF13499	[53]
		Glyma.06G207900	Glycine-rich protein	PF07173	[54]
		Glyma.06G206800	transducin family protein/WD-40 repeat family protein	PF00400, PF12341	[50]
Gm06_38404808	6	Glyma.06G239026	Disease resistance protein (TIR-NBS-LRR class) family	PF13676, PF00931	[57]
		Glyma.06G239500	UDP-glucosyl transferase 72E1	PF00201	[52]
		Glyma.06G238700	Phytosylfokine- α receptor 2	PF08263, PF00069	[58]
Gm09_4708504	9	Glyma.09G051600	Ubiquitin-protein ligase 1		[59]
		Glyma.09G052400	PLATZ transcription factor family protein	PF04640	[60]
		Glyma.09G052700	K+ uptake permease 11	PF02705	[45]
		Glyma.09G056400	Disease resistance protein (TIR-NBS-LRR class), putative	PF13676, PF00931	[57]
		Glyma.09G057500	Transducin family protein/WD-40 repeat family protein	PF00400	[50]
		Glyma.09G057800	Pumilio 5	PF00806	[46]
Gm10_48178692	10	Glyma.10G250000	Leucine-rich repeat (LRR) family protein	PF08263, PF12819	[61]
		Glyma.10G252200	Chaperone DnaJ-domain superfamily protein	PF00226	[48]
		Glyma.10G257600	RING/U-box superfamily protein	PF12906	[62]
		Glyma.10G257700	Transducin/WD40 repeat-like superfamily protein	PF00400	[50]
		Glyma.10G257900	Zinc-finger protein 1	PF13912	[63]
		Glyma.10G258800	Leucine-rich repeat (LRR) family protein	PF08263, PF13855	[61]
		Glyma.10G260000	Pumilio 9	PF00806	[46]
Gm12_34424219	12	Glyma.12G171232	Transducin/WD40 repeat-like superfamily protein	PF00400	[50]
		Glyma.12G173300	RING/U-box superfamily protein	PF12906	[62]
		Glyma.12G173800	FASCICLIN-like arabinogalactan-protein 11	PF02469	[64]
		Glyma.12G175351	Zinc finger (C3HC4-type RING finger) family protein	PF17123, PF14624	[63]
		Glyma.12G172800	PDI-like 1–4	PF13899, PF13848	[65]
Gm13_40079851	13	Glyma.13G310100	WRKY family transcription factor	PF03106	[66]
		Glyma.13G311000	FASCICLIN-like arabinogalactan protein 17 precursor	PF02469	[64]
		Glyma.13G311300	Ubiquitin-specific protease 8	PF13423	[67]
		Glyma.13G312400	Chaperone DnaJ-domain superfamily protein	PF00226	[48]
		Glyma.13G313400	CLAVATA3	IPR039618	[68]
		Glyma.13G314800	Leucine-rich repeat (LRR) family protein	PF13855	[61]
		Glyma.13G317200	WIP domain protein 3	PF13912	[69]
Gm15_12688260	15	Glyma.15G152400	Disease resistance protein (TIR-NBS-LRR class) family	PF13676, PF00931	[57]
		Glyma.15G155900	Transducin/WD40 repeat-like superfamily protein	PF00400	[50]
		Glyma.15G155400	Fasciclin-like arabinogalactan family protein	PF02469	[64]
		Glyma.15G155500	RING/U-box superfamily protein	PF13920	[62]
		Glyma.15G155600	Leucine-rich repeat transmembrane protein kinase	PF08263, PF00560, PF00069, PF13855	[61]
		Glyma.15G154200	CW-type zinc finger	PF07496	[70]
		Glyma.15G154000	Cullin 1	PF10557, PF00888	[71]
Gm16_31759458	16	Glyma.16G152400	RING/U-box superfamily protein	PF13639	[62]
		Glyma.16G153100	Transducin family protein/WD-40 repeat family protein		[50]
		Glyma.16G156100	Leucine-rich repeat transmembrane protein kinase family protein	PF00560, PF00069, PF13855	[61]
		Glyma.16G156400	C2H2 and C2HC zinc fingers superfamily protein	PF13912	[63,72]
		Glyma.16G157200	Ubiquitin carboxyl-terminal hydrolase family protein	PF11955	[67]
		Glyma.16G159100	Disease resistance protein (TIR-NBS-LRR class) family	PF13676, PF00931	[57]
		Glyma.16G159500	Disease resistance protein (TIR-NBS-LRR class), putative	PF00931	[57]
		Glyma.16G155400	Acyl-CoA N-acyltransferases (NAT) superfamily protein	PF00583	[73]
		Glyma.16G031600	NB-ARC domain-containing disease resistance protein	PF00931	[74]
Gm16_3302971	16	Glyma.16G035000.1	Ubiquitin-conjugating enzyme 5	PF00179	[67]
		Glyma.16G033900	Disease resistance protein (TIR-NBS-LRR class), putative	PF07725	[57]
				PF13676	
				PF00931	
		Glyma.16G034400	Transducin/WD40 repeat-like superfamily protein	PF04564	[50]
		Glyma.16G035800	DnaJ/Hsp40 cysteine-rich domain superfamily protein	IPR036410	[49]
		Glyma.16G038500	Cupredoxin superfamily protein	PF02298	[75]
		Glyma.16G031400	WRKY DNA-binding protein 56	PF03106	[66]
Gm16_34649045	16	Glyma.16G181700	C2H2-like zinc finger protein	PF13912	[72]
		Glyma.16G182900	Disease resistance family protein/LRR family protein	PF13855, PF08263	[76]
Gm17_8014133	17	Glyma.17G101800	RING/U-box superfamily protein	PF13639	[62]
		Glyma.17G106000	Ubiquitin-like superfamily protein	PF00240	[67]
		Glyma.17G106300	PLATZ transcription factor family protein	PF04640	[60]
		Glyma.17G107100	Ubiquitin-associated (UBA)/TS-N domain-containing protein	PF02148	[67]
		Glyma.17G107400	Leucine-rich repeat family protein	PF13855	[61]
		Glyma.17G109100	Ubiquitin-specific protease 22	PF02148, PF13423	[67]
		Glyma.17G109200	Avirulence-induced gene (AIG1) family protein	PF04548, PF11886	[70]
Gm17_14222127	17	Glyma.17G162200	F-box family protein	PF07734	[77]
		Glyma.17G162100	Myb domain protein 79	PF00249	[78]
		Glyma.17G162400	TTF-type zinc finger protein with HAT dimerization domain	PF05699, PF14291	[79]
		Glyma.17G161700	Acyl-CoA N-acyltransferases (NAT) superfamily protein	PF00583	[80]
Gm19_35502386	19	Glyma.19G102400	CCR-like	PF07207	[81]
		Glyma.19G102300	Pentatricopeptide repeat (PPR) superfamily protein	PF14432, PF01535, PF13041	[82]
		Glyma.19G100900	AP2/B3-like transcriptional factor family protein	PF02362	[83]
		Glyma.19G102800	UDP-Glycosyltransferase superfamily protein	PF00201	[52]
		Glyma.19G103000	RING/U-box superfamily protein	PF00097	[62]
Gm19_44916522	19	Glyma.19G186200	C2H2-type zinc finger family protein	PF13912	[72]
		Glyma.19G191100	Zinc finger C-x8-C-x5-C-x3-H type family protein	PF00076, PF00642	[63]
		Glyma.19G180600	Homolog of mammalian P58IPK	PF13371, PF13414, PF00226	[84]
		Glyma.19G180700	FASCICLIN-like arabinogalactan protein 11	PF02469	[64]
		Glyma.19G185100,	Transducin/WD40 repeat-like superfamily protein	PF00400	[50]
		Glyma.19G182151	NRAMP metal ion transporter family protein	PF01566	[85]
		Glyma.19G184600	Thioredoxin superfamily protein	PF00085	[86]
		Glyma.19G188200	PLATZ transcription factor family protein	PF04640	[60]
		Glyma.19G189100	Ubiquitin-protein ligase 7	PF00632	[87]
Gm20_34576213	20	Glyma.20G103300	Prefoldin 5	PF02996	[88]
		Glyma.20G098700	F-box family protein	PF00646	[89]
		Glyma.20G099100	C2H2 and C2HC zinc fingers superfamily protein	PF13912	[72]
		Glyma.20G099800	RING/U-box superfamily protein	PF13639	[63]
		Glyma.20G100500	Leucine-rich repeat (LRR) family protein	PF08263, PF00560, PF13855	[61]
		Glyma.20G100800	SNF2 domain-containing protein/helicase domain-containing protein/zinc finger protein-related	PF00097, PF00271, PF00176	[87]
		Glyma.20G104000	thioredoxin X	PF00578	[90]
		Glyma.20G104200	Tetratricopeptide repeat (TPR)-like superfamily protein	PF13812, PF01535, PF13041	[56]
		Glyma.20G107900	AP2/B3-like transcriptional factor family protein	PF02362	[55]

^a^ Single nucleotide polymorphic marker identity. ^b^ Chromosome number. ^c^ Gene identity. ^d^ Protein family.

**Table 7 genes-14-01271-t007:** Profile of the alleles at the significant SNPs for the highly resistant (mean score less than 0.5) soybean genotypes to red leaf blotch recorded out of the 279 genotypes.

Genotypes **	Gm01_17813710 *	Gm06_20112134 *	Gm16_3302971 *	Gm16_34649045 *	Gm19_44916522 *
UGSOY143	GG	GG	AT	GG	CC
UGSOY148	AG	GG	TT	GG	CC
UGSOY157	AA	GG	TT	GG	TT
UGSOY158	AA	GG	TT	GG	TT
UGSOY165	AA	GG	TT	GG	TT
UGSOY170	AG	GG	TT	GG	CC
UGSOY174	GG	GG	TT	GG	CC
UGSOY183	AG	GG	TT	GG	CC
UGSOY186	AA	GG	TT	GG	TT
UGSOY189	AG	GG	TT	GG	TT
UGSOY190	AA	GG	TT	GG	TT
UGSOY194	AA	GG	TT	GG	TT
UGSOY195	AG	GG	TT	GG	TC
UGSOY197	AG	GG	TT	GG	TT
UGSOY203	AG	GG	TT	GG	TT
UGSOY212	AA	GG	TT	GG	TT
UGSOY214	AA	GG	TT	GG	TT
UGSOY217	AA	GG	TT	GG	TT
UGSOY218	AG	GG	TT	GG	TT
UGSOY228	AG	GG	TT	GG	TT
UGSOY229	AG	GG	TT	GG	TT
UGSOY231	AA	GG	TT	GG	CC
UGSOY232	AA	GG	TT	GG	TT
UGSOY233	AA	GG	TT	GG	TT
UGSOY234	AG	GG	TT	GG	TT
UGSOY235	AA	GG	TT	GG	TT
UGSOY236	AG	GG	AT	GG	TT
UGSOY238	AG	GG	TT	GG	TT
UGSOY240	AA	GG	TT	GG	TT
UGSOY241	AA	GG	TT	GG	CC

* Position of the significant SNPs at chromosome 1 at R4 and R6; 8 at R3, R4, and R5; 16 (3302971) at R1 and R4; 16 (34649045) at R2 and R4; and 19 at R5 and R6 in the soybean genome. ** Codes of soybean genotypes with resistance to red leaf blotch disease.

## Data Availability

Data are unavailable because of privacy or ethical restrictions; however, they may be available upon request.

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
