# Peer review of "Genome-Wide Association Analysis for Resistance to Coniothyrium glycines Causing Red Leaf Blotch Disease in Soybean"

_genes, 2023, doi:10.3390/genes14061271_

Round 1

Reviewer 1 Report

Soybean is an important crop world-wide. Identifying gene resources resistant to red leaf blotch is valuable and meaningful to modern soybean breeding. The writing of this study is good, however, there are critical issues of statistical analysis and result elucidation. So I recommend to major revision. All the comments, questions, and suggestions listed as following:

1.Line 118-132: the authors said they assessed the resistance seven times, and they used AUDPC value to do GWAS. I think this strategy is not good. Currently I did not find the most important information, heritability of AUDPC. I suggest to do GWAS for each time point (you can compare the heritabilities for each point score) or every stage (also heritabilities calculation). Then choose the better way to do GWAS, which will definitely better than single AUDPC. And concerning the resistance, the highest value (severest point) should be the better signal than AUDPC. The authors could simultaneously calculate the heritability of the severest point score, and do GWAS for it.

2. I suggest to use ‘that’ instead of ‘,’ in the paper title.

3. Line 104: Taiwan is not a country, I suggest to change ‘countries’ to ‘location’ or ‘site’, or change ‘Taiwan’ to ‘China’, same change for Table 1 and 4.

4. Line 114: insert ‘times of’ between ‘Two hand’.

5. Line 133: the current formula of AUDPC is not right. Check and correct it.

6. Line 140: 96-well, or 96-cell?

7. Line 152: the font size is different to others.

8. Line 157-158: 1st, you lost the ‘)’ after ’Unification’; 2nd, what are ‘both’ for 1%?

9. Line 174-175: how you set “Bonferroni” and what is the “independent variable”?

10.Table3: 1st, the DF of G*L is not right, should be (279-1)*(2-1)=278; 2nd, the name of third col is not right, should be “MS” or “F-value”; 3rd, insert another row at the bottom as ‘Total’ to count the sum of DF.

11.Table4: Current table less information, I suggest to list the BLUE/BLUP value for those genotypes. Another better solution, use a boxplot-like figure to show distribution of different origin and mark out those top 30 genotypes with color or special signs.

12.Line 192-195: ‘DArTseq’ and ‘DArT-seq’, unify them.

13.Table5: MAF should be within 0~0.5, so the col of ‘MAF’ is not right. Line 198-200 expression is right.

14.Line 215 and 220: the three ‘plots’ should be ‘plot’ since there was only one Q-Q plot in Figure 2.

15.Line 217: delete the ‘analysis’, and use GWAS to simplify ‘genome-wide association study’.

16.Table6: 1st, delete ‘describing different genomic regions’ in the title; 2nd, how you set REF and ALT, I did not find the instruction; 3rd, keep 1 or 2 decimals for col of ‘Effect’ and ‘SE’, more make nonsense.

17.Figure 2: 1st, ‘he red’ should be ‘The red’; 2nd, ‘negative log base 10’ and ‘negative base 10 logarithm’, which one is better? 3rd, ‘the 95% confidence interval’ is not right, should be ‘threshold of Bonferroni method at α=0.05 (I’m not sure how big value you set)’.

18.Figure 3: 1st, the authors should do t-test or LSD from ANOVA for those two or three genotypes to express the degree of significance. 2nd, we often mention MAF and generally delete MAF < 0.05, just because the comparison is seriously unbalanced when MAF < 0.05. However, almost all except C look quite unbalanced, GA in A, TA in B, TC in D, and CT&TT in E, the ratio lower than 0.05, it is unfair to compare each of them with its related counterpart for degree of significance.

19.Line 292: change ‘reported in’ to ‘of’.

20.Line 315: ‘an’ to ‘a’.

21.Line 367: add () for 2008.

Author Response

Dear Reviewer

We acknowledge your contribution  to improve the quality of the manuscript entitled “Genome-wide association analysis for the resistance to Coniothyrium glycines that causing red leaf blotch disease in soybean” submitted to Genes journal. All the comments were considered and corrections made as it’s indicated in the attached point-by-point response document . Thanks for your shares for improving this research

Reviewer 2 Report

Comments:

1.       For the genotyping, how many data/subreads generated for each accession?

2.        Please describe more clearly for candidate genes identification.

3.       The author should add more detailed information about how they obtained the genotype, such as sequencing library construction, SNP calling and filtering process, etc. For the final genotype, did they filter the data by missing ratio?

4.        The author should describe more about the genotype, such as density, distribution across the genome etc.

5.       What’s the criterion for the subpopulation classification? How many accessions are admixture?

6.       Abstract needs redrafting. How many SNPs were used for conducting GWAS? Information on major putative genes may be provided.

7.       In the introduction section content from line No. 50 to 57 need to be shorten and modified. Message is not clear

8.       What does it mean by season 2021 B and 2022A .It is not clear in for the general readers.

9.       Please cite a valid reference where it has been clearly mentioned that both the evaluation cites were the hot spots for disease occurrence.

10.   .For GWAS results, I suggest the authors to perform haplotype analysis, homologous gene analysis, GO enrichment analysis, etc

11.   1mntion about the broad sense heritability  estimates of the resistance in the results and discussion section.

Author Response

Dear Reviewer

We acknowledge your contribution to improve the quality of the manuscript entitled “Genome-wide association analysis for the resistance to Coniothyrium glycines that causing red leaf blotch disease in soybean” submitted to Genes journal. All the comments were considered and corrections made as it’s indicated in this point-by-point response. Thanks for your shares for improving this research

Round 2

Reviewer 1 Report

The R1 looks much better than the original. Most of my suggestions and questions were corrected or clarified. However, there were still many items need to be corrected or optimized:

1. Line 99: Change ‘Taiwan’ to ‘China’.

2. Line 141: environment effect (E) was lost in the formula, complement it.

3. Line 148-150: 1st, you lost the ‘)’ after ’Unification’; 2nd, what are ‘both’?

4. Line 169-170: how you set “Bonferroni” and what is the “independent variable”?

5. Line 186-188: you should elucidate which stage (R?) was plotted for this Figure.

6.Line 193: delete ‘-’ within ‘DArT-seq’.

7. Table 5: 1st, delete ‘describing different genomic regions’ in the title; 2nd, Col 4-7 were not in right manner, adjust them.

8.Figure 3(Line 239-243): 1st, ‘he red’ should be ‘The red’; 2nd, ‘the 95% confidence interval’ is not right, should be ‘threshold of Bonferroni method at α=0.05 (I’m not sure how big value you set, that just what I asked at above Point 4)’.

9.Figure 4&5: 1st, the authors should do t-test or LSD from ANOVA for those two or three genotypes to express the degree of significance. 2nd, combine the two figures into one figure, really not necessary to set so wide for each box, narrow down them and you can combine these two figures easily.

Author Response

Dear Reviewer

We acknowledge your contribution to improve the quality of the manuscript entitled “Genome-wide association analysis for the resistance to Coniothyrium glycines that causing red leaf blotch disease in soybean” submitted to Genes journal. All the comments were considered and corrections made as it’s indicated in the attached point-by-point response report. Thanks for your shares for improving this research
